# Sulforaphane Target Protein Prediction: A Bioinformatics Analysis

Francisco Alejandro Lagunas-Rangel 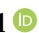

Department of Genetics and Molecular Biology, Centro de Investigación y de Estudios Avanzados del Instituto Politécnico Nacional, Av. Instituto Politécnico Nacional 2508, San Pedro Zacatenco, Gustavo A. Madero, Mexico City 07360, Mexico; francisco.lagunas@cinvestav.mx

**Abstract:** Sulforaphane, a phytochemical found in cruciferous vegetables and various nutraceutical foods, plays a crucial role in promoting well-being and combating various diseases. Its remarkable effects are due to its intricate interactions with a wide range of proteins, some of which remain unidentified. In this study, taking advantage of bioinformatics tools for protein target prediction, we identified 11 proteins as potential targets of sulforaphane. Due to its biological relevance and their correlation with transcriptomic changes observed in sulforaphane-treated cells, the possible interaction between sulforaphane and nicotinamide phosphoribosyltransferase (NAMPT) was further investigated. A docking analysis suggested that sulforaphane is strategically positioned at the entrance of the channel through which substrates enter, thus bypassing the active site of the enzyme. By forming hydrogen bonds with residues K189, R349, and S275, sulforaphane establishes a linkage with NAMPT. Dynamic molecular analyses further corroborated these observations, illustrating that these bonds allow sulforaphane to associate with NAMPT, mimicking the behavior of a NAMPT activator (NAT), a known activating compound of this enzyme. This collective evidence suggests that sulforaphane may activate NAMPT, providing valuable insights into a possible mechanism underlying its diverse biological effects.

**Keywords:** phytochemical; NAMPT activation; NAD+; NRF2; oxidative stress response

## 1. Introduction

Sulforaphane (1-isothiocyanate-4-(methylsulfinyl)butane), a naturally occurring isothiocyanate found in cruciferous vegetables such as cabbage, cauliflower and kale, is especially abundant in broccoli sprouts [1]. This compound exhibits two optical isomers (R [Rectus → Latin = Right] and S [Sinister → Latin = Left]) (Figure 1A,B) due to its asymmetric sulfur atom, both of which are present in broccoli sprouts [2]. In particular, the research highlights the superior biological activities exhibited by the R-enantiomer of sulforaphane in comparison to the S-enantiomer [3,4].

Within plant structures, sulforaphane exists as an inactive precursor known as glucoraphanin (Figure 1C), which undergoes rapid conversion to its active form by the plant enzyme myrosinase [5]. This transformation takes place when vegetables are bitten and chewed, resulting in the release of glucoraphanin and myrosinase. However, it is crucial to recognize that myrosinase is susceptible to destruction during cooking, steaming, or microwaving, resulting in decreased sulforaphane levels in cooked vegetables [6,7]. After ingestion, sulforaphane undergoes metabolic transformation via the mercapturic acid pathway, involving its conjugation with glutathione. This initial conjugation paves the way for further biotransformation processes, leading to the generation of various metabolites such as sulforaphane-cysteine and sulforaphane-N-acetylcysteine [1,8]. Sulforaphane elimination is characterized by a prolonged terminal phase. For example, in humans, the maximum plasma concentration was recorded at 3 h after administration of 200 μM sulforaphane, with an elimination half-life of $1.9 \pm 0.4$ h [9].

**Figure 1.** Chemical structures of the sulforaphane enantiomers (**A,B**) and its precursor, glucoraphanin (**C**).

Sulforaphane has been the subject of much research, attracting considerable attention in the medical field due to its various health benefits [1]. The therapeutic potential of sulforaphane is underscored by the participation of nearly 100 clinical trials, each at different stages, that systematically explore its potential health benefits in a variety of conditions. These trials cover areas such as autism, schizophrenia, depressive disorders, aging and specific types of cancer [8]. Notably, numerous studies are dedicated to unraveling the intricate mechanisms through which sulforaphane exerts its therapeutic effects [10–13]. The reported effects of sulforaphane span a spectrum of physiological and biochemical alterations, including its influence on oxidative stress, enhancement of antioxidant capacity, elimination of cancer cells and reduction of neuroinflammation [1,8]. These multifaceted actions are attributed to the modulation of various proteins by sulforaphane. Significantly, activation of the NRF2 (nuclear factor erythroid 2-related factor 2) pathway has garnered substantial attention as the fundamental mechanism through which sulforaphane exerts its effects [14]. However, it is noteworthy that sulforaphane extends its influence beyond this pathway, affecting other signaling pathways [15,16], metabolic pathways [17] and gene transcription processes [18,19].

Despite notable advances, the complete catalog of sulforaphane-interacting proteins remains elusive. Moreover, it is crucial to recognize that the effects of this compound are dose dependent, suggesting an affinity-dependent binding to target proteins [20]. This dose-dependent nature adds complexity to the understanding of sulforaphane's actions, emphasizing the nuanced interaction between the compound and its molecular targets.

In the last decade, bioinformatics has undergone significant advances, transforming our ability to predict molecular targets for new drugs and repurpose existing compounds for new therapeutic applications [21,22]. This transformation involves the integration of high-throughput data and machine learning techniques, enabling the retrospective identification of key proteins implicated in drug-induced transcriptomic and proteomic changes [23,24]. Furthermore, approaches based on protein–protein interaction networks, drug–target networks, and disease–gene networks have contributed to improving our understanding of protein–protein interaction mechanisms [25–27].

The intricate interaction between sulforaphane and its target proteins is a captivating field of research with potential for revolutionary discoveries. The integration of bioinformatics tools is of great help to significantly advance this field, facilitating the more efficient and accurate screening of candidate proteins. Therefore, the main objective of this work was to identify new potential protein targets of sulforaphane in order to deepen our understanding of its biological functions and therapeutic potential.

## 2. Materials and Methods

### 2.1. Prediction of Protein Targets

The identification of potential sulforaphane protein targets was conducted utilizing SwissTargetPrediction [28], SuperPred [29] and TargetNet [30], employing default parameters across all platforms. Only proteins that exhibited predictions on all three servers were considered for further analysis.

### 2.2. RNA Sequence Analysis

Publicly available sequencing data from the Gene Expression Omnibus (GEO) database [31], specifically GEO accession number GSE141740 [14], were used. The data were from WT 9–12 cells subjected to treatment with 0.1% DMSO or 10 μM sulforaphane for 24 h, followed by sequencing on the Illumina NovaSeq 6000 platform. WT 9–12 cells are epithelial cell lines that have been immortalized with SV40 large T antigen. The quality of the raw reads was assessed using the FastQC toolkit, and subsequent removal of low-quality reads and adapters was performed with Trimmomatic [32]. To align the raw reads to the GRCh38.p5 reference genome (Ensembl version 84), HISAT2 [33] was employed. Expression levels of each gene were quantified using featureCounts [34] based on Ensembl annotation version 84. Poorly expressed genes, defined as those with no more than one count per million reads (1CPM) in at least two samples within each dataset, were excluded from further analysis. The raw counts were then normalized and used to perform differential expression analysis with Limma [35]. In addition, edgeR [36] was used to validate the results obtained with Limma. Only genes that showed consensus of overexpression or underexpression across both software packages were considered. Genes with a log2 fold change (log2fc) greater than 1.0 and an adjusted *p*-value (padj) less than 0.05 were classified as upregulated, whereas those with a log2fc less than 1.0 and a padj less than 0.05 were considered downregulated.

### 2.3. Functional Enrichment Analysis

Based on the consensus of genes identified as upregulated and downregulated with sulforaphane in the RNA-seq analysis; an analysis was performed to identify enriched biological pathways and processes. This analysis was performed using the g:Profiler server [37] with predefined parameters. Furthermore, the investigation was extended by exploring possible sulforaphane-modified functional associations by delving into the genes identified as overexpressed or underexpressed in the RNA-seq analysis. This exploration was carried out using the STRING database [38,39], applying predetermined parameters.

### 2.4. Docking Analysis

The crystallized structure of NAMPT (2E5B) [40] was obtained from the Protein Data Bank (PDB) [41], while the structure of sulforaphane (ZINC3875035) was obtained from the ZINC20 database (https://zinc20.docking.org) (accessed on 15 December 2023) [42]. Docking studies were performed using SwissDock [43] with default parameters. To facilitate comparisons, the NAMPT structure bound to the NAMPT activator (NAT) (7ENQ) [44] was used. Figure 2 shows the chemical structure of NAT.

**Figure 2.** Chemical structure of NAT.

## 2.5. Molecular Dynamics (MD) Simulation

MD simulations were conducted using the s [45]. Hydrogen atoms for proteins were incorporated with the *tleap* module based on the *ff14SB* force field [46]. Force field parameters for all candidates were generated using the Antechamber module using the AM1-BCC loading model [47]. The systems were immersed in TIP3P water [48] and chloride ions were introduced for neutralization. Energetic minimization involved 4000 maximum descent steps followed by 1000 conjugate gradient steps. The systems were subjected to heating in the NPT ensemble, moving from 0 to 300 °K with weak restraints (simulations were performed with a time step of 2 femtoseconds (fs) for all systems, using a strength parameter κ = 10) within the proteins for a period of 30 picoseconds (ps). Subsequently, 100 ns MD simulations were performed on the NVT array, maintaining a temperature of 300 °K. Temperature control was achieved with the Langevin thermostat, and pressure was regulated using the anisotropic Berendsen barostat. Root mean square deviations (RMSD) and root mean square fluctuations (RMSF) were calculated with respect to the initial conformation. To determine the binding free energies ($\Delta G_{Binding}$), 100 snapshots were extracted every 100 ps from the last stable MD trajectory of 10 ns, and the analysis was performed with MM-PBSA.py [49].

## 3. Results

### 3.1. Predicted Targets of Sulforaphane

To identify new protein targets of sulforaphane, three bioinformatics platforms (SwissTargetPrediction, SuperPred and TargetNet) were used, each using different algorithms to predict the most likely targets of this compound. The SwissTargetPrediction search identified 102 targets, SuperPred generated 115 targets and TargetNet revealed 623 targets. Through a comprehensive analysis of the candidate proteins, 11 proteins were consistently present in all three lists, establishing them as the most likely targets of sulforaphane (Table 1).

**Table 1.** Consensus sulforaphane targets predicted using SwissTargetPrediction, SuperPred and TargetNet.

| Target | Gene | Uniprot ID | ChEMBL ID | Target Class |
|---|---|---|---|---|
| Macrophage migration inhibitory factor | MIF | P14174 | CHEMBL2085 | Tautomerase |
| Cytochrome P450 19A1 | CYP19A1 | P11511 | CHEMBL1978 | Aromatase |
| Nicotinamide phosphoribosyltransferase | NAMPT | P43490 | CHEMBL1744525 | Transferase |
| Poly [ADP-ribose] polymerase-1 | PARP1 | P09874 | CHEMBL3105 | Transferase |
| P2X purinoceptor 7 | P2RX7 | Q99572 | CHEMBL4805 | Ligand-gated ion channel |
| Glycogen synthase kinase-3 β | GSK3B | P49841 | CHEMBL262 | Kinase |
| Cyclin-dependent kinase 2 | CDK2 | P24941 | CHEMBL301 | Kinase |
| Cathepsin L | CTSL | P07711 | CHEMBL3837 | Protease |
| Adenosine A1 receptor | ADORA1 | P30542 | CHEMBL226 | Family A G protein-coupled receptor |
| Adenosine A2a receptor | ADORA2A | P29274 | CHEMBL251 | Family A G protein-coupled receptor |
| Monoamine oxidase A | MAOA | P21397 | CHEMBL1951 | Oxidoreductase |

Among the 11 proteins identified, previous evidence of interaction with sulforaphane exists for MIF (Macrophage migration inhibitory factor) [50], CDK2 (Cyclin-dependent kinase 2) [51], GSK3β (Glycogen synthase kinase-3β) [52–55] and PARP1 (Poly [ADP-ribose] polymerase 1) [56,57]. However, no previous evidence of interaction with sulforaphane was found for the remaining seven proteins (CYP19A1 [Cytochrome P450 19A1], NAMPT [Nicotinamide phosphoribosyltransferase], P2RX7 [P2X purinoceptor 7], CTSL [Procathep-

sin L], ADORA1 [Adenosine receptor A1], ADORA2A [Adenosine receptor A2A], and MAOA [Monoamine oxidase A]).

### 3.2. Changes in Transcriptomic Profile Caused by Sulforaphane

To deepen the knowledge of possible new protein targets of sulforaphane, RNA sequencing data (RNA-seq) obtained from cells treated with this compound were analyzed. Sequences were obtained from public databases and a comprehensive differential expression analysis was performed using the bioinformatics packages Limma and edgeR. Only genes consistently identified as overexpressed or underexpressed by both packages were considered. This rigorous approach yielded a total of 188 overexpressed and 207 underexpressed genes (Figure 3A and Supplementary Data). Notably, Table 2 shows the list of the 10 upregulated and downregulated genes.

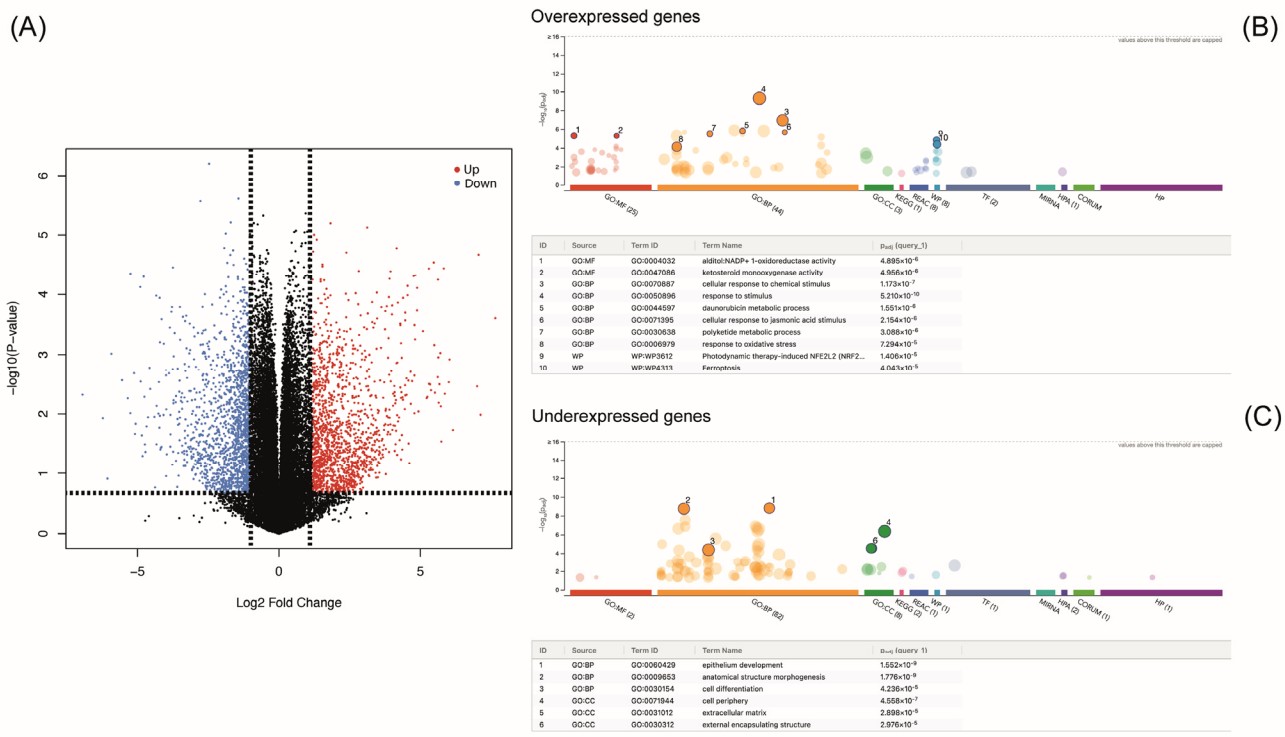

**Figure 3.** RNA-seq and functional enrichment analysis of sulforaphane-treated cells. Volcano plot of RNA-seq analysis showing overexpressed and underexpressed genes in sulforaphane-treated cells (**A**). Functional enrichment analysis performed with overexpressed (**B**) and underexpressed (**C**) genes identified in RNA-seq analysis.

This integrative analysis of RNA-seq data provides a more comprehensive view of the cellular response to sulforaphane treatment. The identified genes, which show altered expression levels, present potential candidates for molecular targets of sulforaphane, providing a basis for further exploration and validation in elucidating the biological effects of the compound. Significantly, NAMPT, identified as a potential target of sulforaphane in this study, was shown to be overexpressed (1.03-fold increase, padj = 0.042). No other proteins predicted as potential targets of sulforaphane showed significant over- or underexpression.

**Table 2.** Top 10 most upregulated and downregulated genes with sulforaphane.

| Upregulated | | | | | | |
|---|---|---|---|---|---|---|
| **Gene** | **logFC** | **AveExpr** | **t** | ***p* Value** | **adj.P.Val** | **B** |
| LINC00536 | 7.64755127 | −3.0149304 | 21.9961465 | 0.00024888 | 0.03788923 | 0.36312489 |
| IL1R2 | 7.05727373 | −3.3119119 | 38.9200915 | $2.15 \times 10^{-5}$ | 0.0314058 | 1.11208622 |
| BBOX1-AS1 | 5.82578022 | −3.9274667 | 32.9561069 | $3.65 \times 10^{-5}$ | 0.0314058 | 0.96501409 |
| PLEKHS1 | 5.70468219 | −3.9959986 | 13.8792101 | 0.00094197 | 0.04820704 | −0.1886544 |
| AC105177.1 | 5.62076293 | −4.0337932 | 28.649169 | $5.71 \times 10^{-5}$ | 0.0314058 | 0.88104941 |
| LAMC3 | 5.49154119 | −4.1016187 | 15.2901064 | 0.00069452 | 0.04291078 | −0.0618761 |
| TNIP3 | 5.4717553 | −4.1066281 | 35.425051 | $2.90 \times 10^{-5}$ | 0.0314058 | 0.95355416 |
| AC020571.1 | 5.30127359 | −4.1922612 | 32.8798182 | $3.68 \times 10^{-5}$ | 0.0314058 | 0.90642193 |
| AKR1C3 | 5.09234014 | 5.97048189 | 18.5451872 | 0.00040802 | 0.04045807 | 0.71525274 |
| LINC00565 | 4.88357902 | −4.402276 | 25.9281408 | $7.85 \times 10^{-5}$ | 0.03315288 | 0.73875417 |
| **Downregulated** | | | | | | |
| **Gene** | **logFC** | **AveExpr** | **t** | ***p* Value** | **adj.P.Val** | **B** |
| DCN | −5.9219991 | −3.5794442 | −13.758792 | 0.00096589 | 0.04824652 | −0.1635847 |
| MARCOL | −5.2456221 | −3.9144047 | −30.897349 | $4.49 \times 10^{-5}$ | 0.0314058 | 0.94001916 |
| NKX6-1 | −4.9009606 | −4.0843505 | −26.237668 | $7.56 \times 10^{-5}$ | 0.03284137 | 0.82265699 |
| TRPC6 | −4.3352769 | −4.3671491 | −23.63015 | 0.00010555 | 0.03426141 | 0.69136095 |
| AC138866.2 | −4.2150679 | −4.4272963 | −23.013573 | 0.00011483 | 0.03426141 | 0.65751575 |
| LINC02208 | −4.2045923 | −4.4300338 | −14.309786 | 0.00062259 | 0.04291078 | 0.0378538 |
| AC016026.1 | −4.1467169 | −4.4601074 | −17.831801 | 0.0002586 | 0.03811126 | 0.43408431 |
| AC106738.1 | −4.0685678 | −4.5062475 | −13.933678 | 0.00066005 | 0.04291078 | 0.0053519 |
| AC008687.6 | −3.8557043 | −4.6052296 | −16.118801 | 0.00035632 | 0.03919762 | 0.28826772 |
| AL121761.2 | −3.8557043 | −4.6052296 | −16.118801 | 0.00035632 | 0.03919762 | 0.28826772 |

### 3.3. Functional Enrichment Analysis

Following RNA-seq analysis, a comprehensive functional enrichment analysis was performed using the g:Profiler server. For overexpressed genes, enriched categories included response to stimuli (GO:0050896, padj = $5.210 \times 10^{-10}$), cellular response to chemical stimuli (GO:0070887, padj = $1.173 \times 10^{-7}$), alditol:NADP+ 1-oxidoreductase activity (GO:0004032, padj = $4.895 \times 10^{-6}$), ketosteroid monooxygenase activity (GO:0047086, padj = $4.956 \times 10^{-6}$), daunorubicin metabolic process (GO:0044597, padj = $1.551 \times 10^{-6}$), cellular response to jasmonic acid stimulus (GO:0071395, padj = $2.154 \times 10^{-6}$), polyketide metabolic process (GO:0030638, padj = $3.088 \times 10^{-6}$), oxidative stress response (GO:0006979, padj = $7.294 \times 10^{-5}$), photodynamic therapy-induced NFE2L2 (NRF2) survival signaling (WP:WP3612, padj = $1.406 \times 10^{-5}$), and ferroptosis (WP:WP4313, padj = $4.043 \times 10^{-5}$) (Figure 3B). Conversely, underexpressed genes targeted epithelial development (GO:0060429, padj = $1.552 \times 10^{-9}$), morphogenesis of anatomical structures (GO:0009653, padj = $1.776 \times 10^{-9}$), cell differentiation (GO:0030154, padj = $4.236 \times 10^{-5}$), cell periphery (GO:0071944, padj = $4.558 \times 10^{-7}$), extracellular matrix (GO:0031012, padj = $2.898 \times 10^{-5}$) and external encapsulating structure (GO:0030312, padj = $2.976 \times 10^{-5}$) (Figure 3C).

To further the analysis, a protein–protein interaction network functional enrichment analysis was carried out. The results of this analysis resulted in the construction of a network that agrees well with the functional enrichment results (Figures 4 and 5).

In particular, annotated keywords associated with overexpressed genes in the network prominently include NADP (strength 0.92, false discovery rate 0.00038), whereas protein domains are predominantly linked to the NADP-dependent oxidoreductase domain superfamily (strength 1.62, false discovery rate 0.00067). Furthermore, a substantial portion of the overexpressed genes were observed to play a regulatory role in cytokine signaling in the immune system (strength 0.35, false discovery rate 0.0109). In contrast, protein–protein interaction network functional enrichment analysis for sulforaphane-underexpressed genes revealed dysregulation in extracellular matrix organization, intracellular signaling by sec-

ond messengers (strength 0.25, false discovery rate 0.0098) and cell differentiation (strength 0.55, false discovery rate $2.07 \times 10^{-6}$).

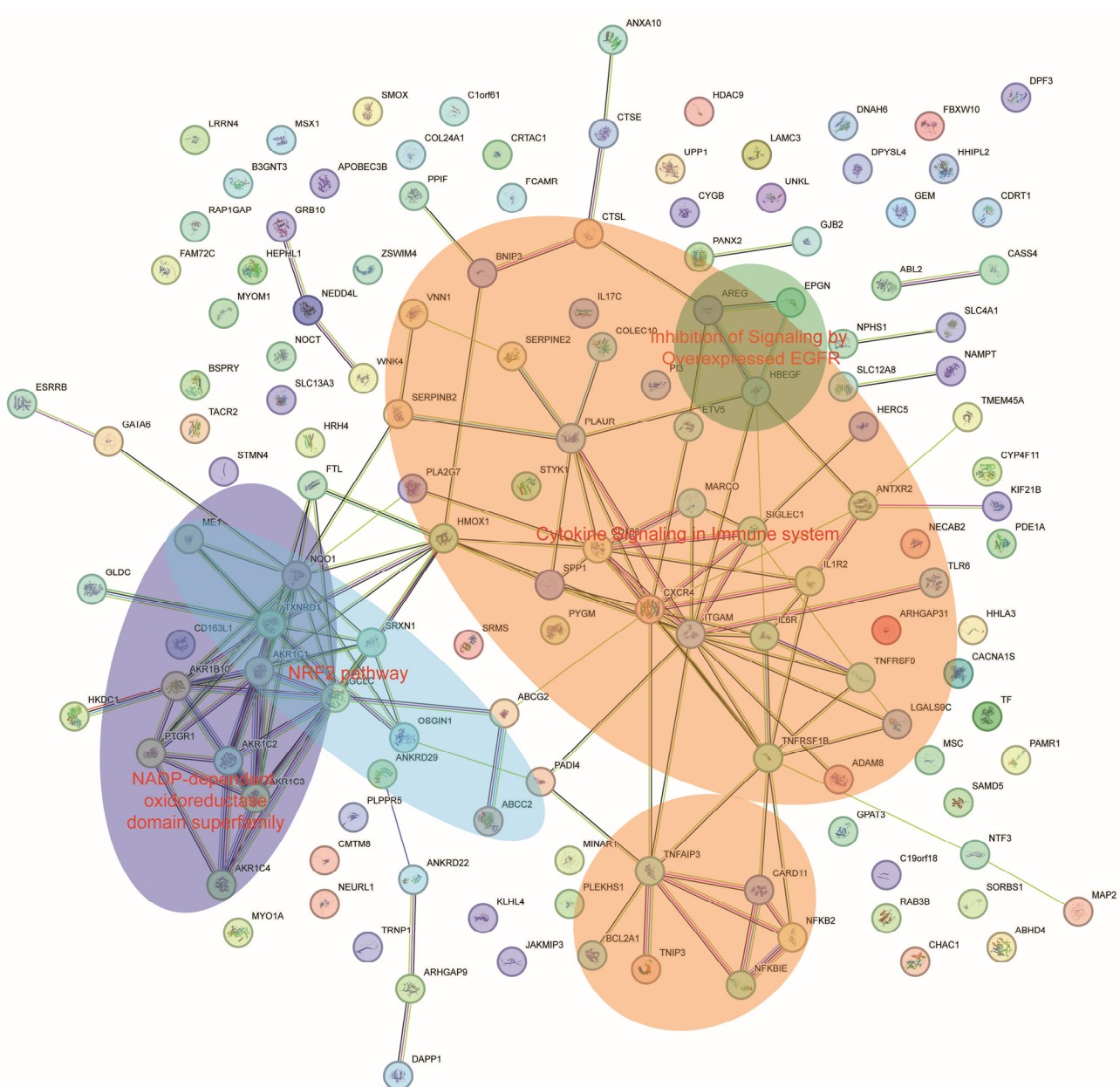

**Figure 4.** Protein–protein interaction network functional enrichment analysis for sulforaphane-overexpressed genes.

Given these results, NAMPT was considered to be the main candidate that could interact with sulforaphane and be associated with these transcriptional changes. Thus, subsequent sections of this work focused on a detailed exploration of this protein.

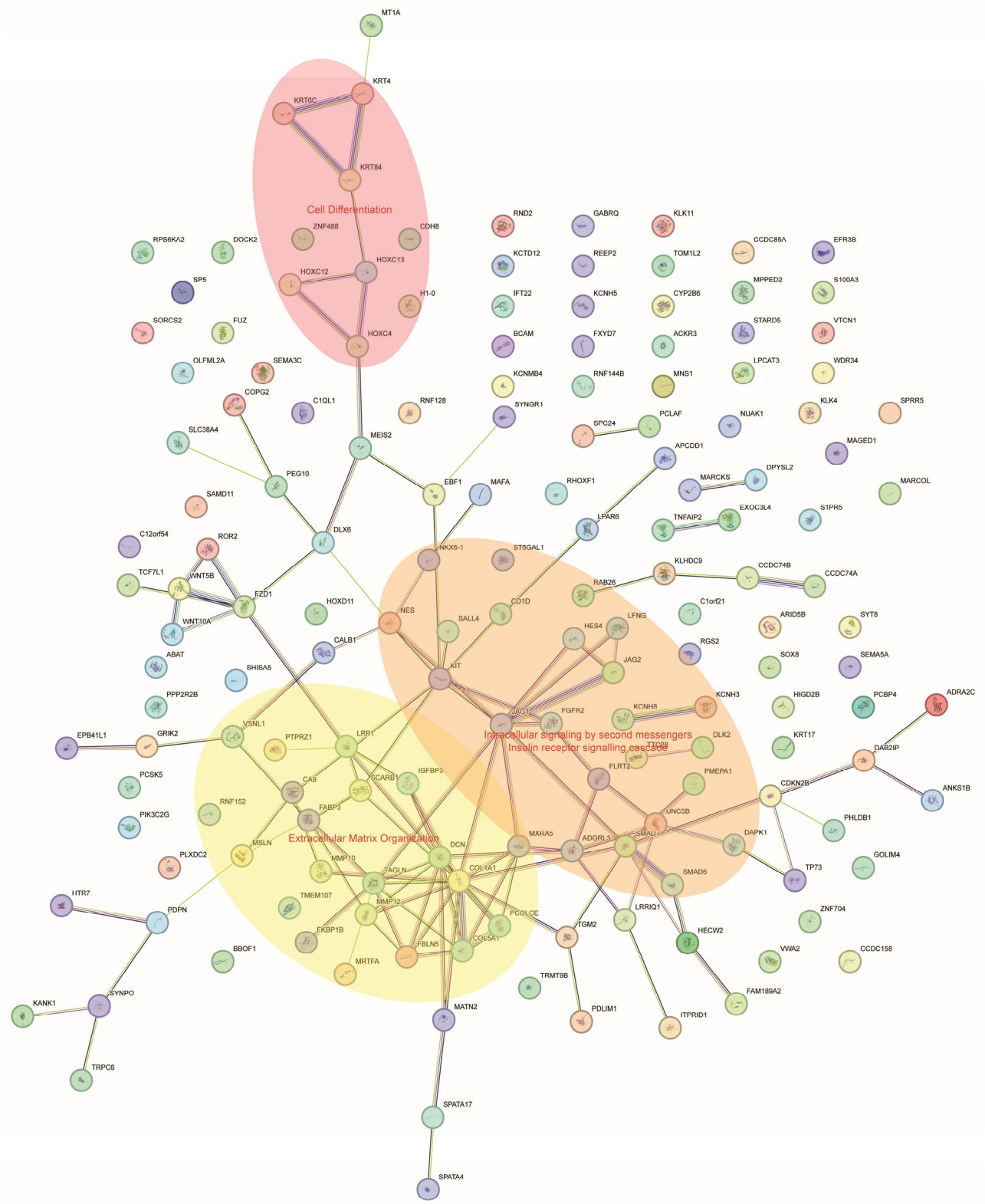

**Figure 5.** Protein–protein interaction network functional enrichment analysis for sulforaphane-underexpressed genes.

### 3.4. Docking Analysis Predicts That Sulforaphane Interacts with NAMPT

The crystallized structure of NAMPT, resolved at a resolution of 2.00 Å, provided the foundation for this investigation. On this structural basis, a meticulous docking analysis was performed to examine the interaction dynamics between NAMPT and sulforaphane.

Interestingly, like NAT (Figure 4A,B), a known activating compound of this enzyme, sulforaphane exhibited positioning at one end of the channel responsible for substrate entry, strategically distanced from the active site (Figure 6C,D).

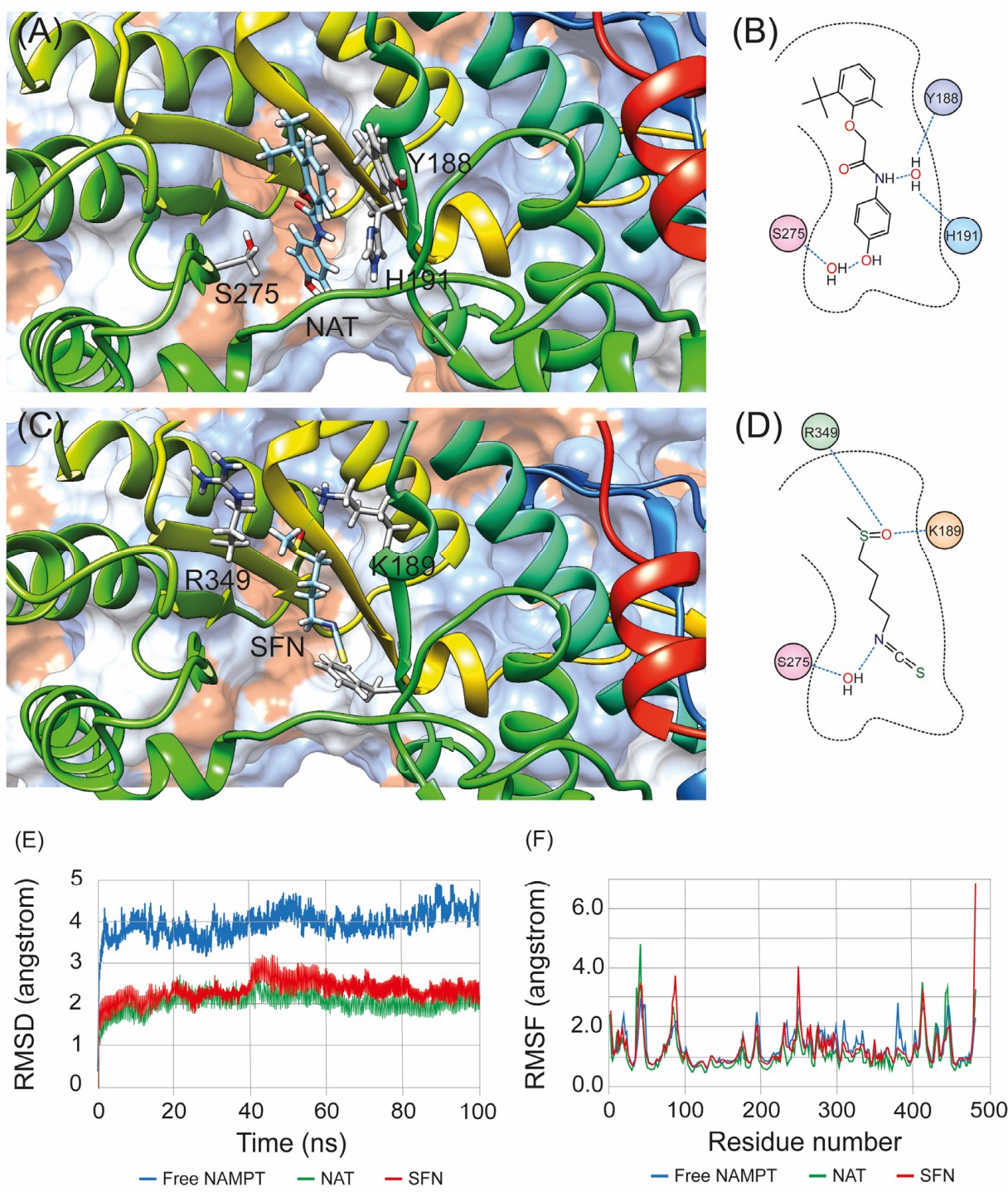

**Figure 6. Sulforaphane (SFN) interacts with NAMPT.** The three-dimensional model shows how NAT interacts with NAMPT (**A**,**B**) and the similar manner in which sulforaphane does (**C**,**D**). Comparative molecular dynamics analysis (**E**,**F**) of NAMPT without ligand (blue lines) and in complex with NAT (green lines) or sulforaphane (red lines). RMSD: root mean square deviations, RMSF: root mean square fluctuation.

This strategic placement ensures that the presence of sulforaphane does not disrupt either the substrates or the enzyme product. Notably, sulforaphane binding did not induce any significant architectural changes in the enzyme, underscoring the non-disruptive nature of this interaction. Sulforaphane was predicted to establish hydrogen bonds between its sulfinyl group, a potent electron-extracting entity, and the electron-donating amino groups of residues K189 and R349 of NAMPT. In addition, it is possible for it to form a hydrogen bond with a water molecule which, in turn, is hydrogen bonded to the hydroxyl group of residue S275 of NAMPT.

The $\Delta G_{Binding}$ calculated for the sulforaphane-NAMPT complex ($-29.96$ kcal/mol) was found to be higher than that for the NAT-NAMPT complex ($-53.69$ kcal/mol), indicating a lower affinity for NAMPT in the case of sulforaphane. Molecular dynamics simulations were used to evaluate the interaction between sulforaphane and NAMPT over a period of 100 ns. The stability of both NAT and sulforaphane within the NAMPT channel was evident throughout the simulation period (Figure 6E). The RMSD plot shows mean deviations ranging from 1.0 to 1.5 Å, with the sulforaphane complex showing slightly higher values. Although the NAT complex reached equilibrium earlier, settling in the initial 50 ns, as is typical for this type of analysis, both complexes demonstrated stability. RMSF analysis revealed comparable fluctuations in both complexes, although more pronounced differences were observed in the sulforaphane-NAMPT complex compared to NAT-NAMPT and ligand-free NAMPT, suggesting greater flexibility (Figure 6F). These observations collectively propose the potential of sulforaphane to act as an allosteric activator of NAMPT, warranting experimental validation of these results.

## 4. Discussion

Sulforaphane, found in abundance in cruciferous vegetables, is emerging as a versatile compound with numerous health benefits when incorporated into the diet. Rigorous research has revealed its promising potential to treat a variety of diseases. In particular, sulforaphane has demonstrated its ability to reverse leptin and insulin resistance, making it a valuable ally in the fight against diabetes and obesity [16,58]. Beyond metabolic health, sulforaphane also shows promise in the treatment of neurodegenerative conditions such as Alzheimer's disease, Parkinson's disease and multiple sclerosis [59]. Moreover, the anticancer properties of sulforaphane have captured great attention in the scientific community [60]. Ongoing studies are delving into the intricate mechanisms through which sulforaphane may contribute to cancer prevention and treatment, shedding light on its potential as a natural compound with profound implications for human health.

Most of the benefits attributed to sulforaphane arise from its activation of the NRF2-mediated antioxidant response [14,60]. However, evidence suggests that this compound interacts with several target proteins, broadening the spectrum of its effects [61,62]. The aim of this study was to identify other target proteins of sulforaphane to better understand its diverse beneficial effects. Taking advantage of target prediction servers with different algorithms, 11 potential sulforaphane-interacting proteins were identified. Evidence of interaction was found for four of them, showing inhibitory effects of sulforaphane on MIF tautomerase activity [50], reduction of CDK2 expression [51], modulation of the GSK3β signaling pathway [53–55], and inhibition of PARP1 activity [56].

After considering the remaining seven proteins for further analysis (CYP19A1, NAMPT, P2RX7, CTSL, ADORA1, ADORA2A and MAOA), attention was given to the NAMPT protein due to its metabolic relevance and correlation with the functional enrichment analysis of overexpressed genes in sulforaphane-treated cells. The analysis highlighted its association with NADP+ 1-oxidoreductase activity, ketosteroid monooxygenase activity and oxidative stress response. These processes are closely related to nicotinamide adenine dinucleotide (NAD+) metabolism [63]. NAMPT catalyzes the limiting step in the mammalian NAD+ salvage pathway, converting nicotinamide and 5′-phosphoribosyl-1-pyrophosphate to nicotinamide mononucleotide (NMN). Subsequently, NMN together with ATP is con-

verted to NAD+ by NMNAT1-3 (nicotinamide/nicotinic acid mononucleotide adenylyl adenylyl-transferases) [64].

In addition to upregulating NAD+, NAMPT also elevates NADP+ and NADPH levels, stimulating oxidative stress systems such as glutathione and thioredoxin [65,66]. This intricate interplay between NAD+ metabolism and cellular redox balance positions NAMPT as a crucial player in the regulation of oxidative stress [66]. The interaction of sulforaphane with NAMPT reveals a potential pathway for modulating cellular redox dynamics, warranting further investigation into its implications for health and disease.

NAT functions as an allosteric activator of NAMPT, strategically positioning itself near the active site of the enzyme. The hydroxyl end of NAT aligns closely with the amide groups of NAM and NMN, resulting in a subtle adjustment of the enzyme's substrate binding affinity for NAM. Specifically, the oxygen atom of the hydroxyl of NAT forms a hydrogen bond with a water molecule and binds to the side chain of residue S275 of NAMPT. In addition, the interaction involves the central amide nitrogen of NAT and Y188 and H191, facilitated by another water molecule [44]. In the docking analysis, sulforaphane was observed to interact with NAMPT by forming hydrogen bonds. Specifically, the sulfinyl group of sulforaphane established hydrogen bonds with the amino groups of residues K189 and R349 of NAMPT. In addition, sulforaphane formed a hydrogen bond with a water molecule, which, in turn, was hydrogen bonded to the hydroxyl group of residue S275 of NAMPT. These interactions closely resemble the crucial role played by residue K189 in driving NAMPT activity, as had already been noted for NAT [44]. Thus, sulforaphane appears to share a similar mechanism of action with NAT in modulating NAMPT function. Notably, the $\Delta G_{Binding}$ value of sulforaphane is higher than that of NAT, indicating a lower affinity. This suggests that although sulforaphane exhibits a comparable mode of action, further research is needed to understand the nuances of its interaction with NAMPT and its potential implications on cellular processes.

## 5. Limitations of the Study

The present study provides exclusively bioinformatically generated data, so it is necessary to emphasize the predictive nature of the results presented. As reiterated in other sections, experimental validation in future studies is imperative. Indeed, these findings could serve as a motivation for research groups around the world, inspiring them to experimentally validate our results and further investigate the mechanisms underlying the beneficial effects induced by sulforaphane.

## 6. Conclusions

The results revealed in this study have allowed us to identify possible protein targets of sulforaphane, with remarkable prediction indicating its ability to activate NAMPT. However, it is crucial to underline that these results require rigorous experimental validation to corroborate their reliability. The completion of these experimental validations will not only validate the computational predictions, but will also provide valuable insights into the intricate molecular mechanisms underlying the interaction between sulforaphane and NAMPT, providing a solid foundation for future studies exploring the therapeutic implications of this interaction.

**Supplementary Materials:** The following supporting information can be downloaded at: https://www.mdpi.com/article/10.3390/app14031052/s1, Supplementary data-RNA-seq analysis.

**Funding:** This research did not receive any specific grant from funding agencies in the public, commercial, or not-for-profit sectors.

**Institutional Review Board Statement:** Not applicable.

**Informed Consent Statement:** Not applicable.

**Data Availability Statement:** The data presented in this study are available on request from the corresponding author.

**Acknowledgments:** The author expresses his gratitude to Helgi B. Schiöth for his invaluable assistance in responding to reviewers' comments on this manuscript.

**Conflicts of Interest:** The author declares no conflicts of interest.

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
