# Peer review of "Sulforaphane Target Protein Prediction: A Bioinformatics Analysis"

_applsci, doi:10.3390/app14031052_

Round 1

Reviewer 1 Report

Comments and Suggestions for Authors

The manuscript entitle “Sulforaphane target protein prediction: A bioinformatics analysis” is well described by the author.

Suggestions and corrections:

Ø  In the abstract, line 15, elaboration of the name of the enzyme NAMPT, Nicotinamide phosphoribosyl-transferase (NAmPRTase or NAMPT) is more appropriate.

Ø  It would be more appropriate to include the structures of R & S-Sulforaphane and its inactive precursor Glocoraphanin compounds in the introduction part. It’s also more informative with the structures of NAT, NAD in the introduction section.

Ø  It would be more appropriate to cite a recent review published in the journal of Molecules in the introduction section.  Otoo, R.A.; Allen, A.R. Sulforaphane’s Multifaceted Potential: From Neuroprotection to Anticancer Action. Molecules 2023, 28, 6902. https://doi.org/10.3390/molecules28196902

Ø  End page numbers are missed in the references 5, 14, 15, 17, 18, 21, 22, 23, 49, 51, 57. Please correct these.

The current manuscript is well described the computational study of the interactions of Sulforaphane with NAMPT enzyme which is closely resembles the interactions of NAT with NAMPT enzyme. The manuscript is acceptable with the above said minor corrections.

Comments on the Quality of English Language

The English language is good.

Author Response

Dear Reviewer,

We express our sincere gratitude for your constructive comments and the time you dedicated to reviewing our manuscript. We have diligently worked to enhance the quality of the manuscript in alignment with your valuable feedback. Please find our responses to your comments below:

The manuscript entitle “Sulforaphane target protein prediction: A bioinformatics analysis” is well described by the author.

Suggestions and corrections:

Ø  In the abstract, line 15, elaboration of the name of the enzyme NAMPT, Nicotinamide phosphoribosyl-transferase (NAmPRTase or NAMPT) is more appropriate.

It was corrected as you requested.

Ø  It would be more appropriate to include the structures of R & S-Sulforaphane and its inactive precursor Glocoraphanin compounds in the introduction part. It’s also more informative with the structures of NAT, NAD in the introduction section.

The chemical structures you mentioned have been incorporated into the new Figures 1 and 2.

Ø  It would be more appropriate to cite a recent review published in the journal of Molecules in the introduction section.  Otoo, R.A.; Allen, A.R. Sulforaphane’s Multifaceted Potential: From Neuroprotection to Anticancer Action. Molecules 2023, 28, 6902. https://doi.org/10.3390/molecules28196902

This reference has been incorporated and is now listed as the current citation 8 in the reference list.

Ø  End page numbers are missed in the references 5, 14, 15, 17, 18, 21, 22, 23, 49, 51, 57. Please correct these.

The page numbers of these references are present, although they deviate from the typical format. Unfortunately, since we have no control over the formatting decisions made by the journals in which the cited articles are published, we cannot modify them. Please be assured that the references cited are accurately cited.

The current manuscript is well described the computational study of the interactions of Sulforaphane with NAMPT enzyme which is closely resembles the interactions of NAT with NAMPT enzyme. The manuscript is acceptable with the above said minor corrections.

Reviewer 2 Report

Comments and Suggestions for Authors

Dear authors, the submitted manuscript is well-conceived, in-depth, informative, and scientifically structured.   However, I would like to add a few suggestions that might be implemented to modify the manuscript for better quality control.

The introduction section could be more comprehensive.

While the study on sulforaphane and its potential interaction with NAMPT is intriguing, it's important to acknowledge certain weaknesses or limitations in the research:

The conclusion that sulforaphane may activate NAMPT is based on computational predictions. Experimental evidence, such as enzymatic assays or in vitro studies, is necessary to confirm this activation and to understand the extent of its impact.

The study primarily focuses on molecular interactions and may lack direct clinical evidence. To establish the practical significance of sulforaphane's interaction with NAMPT, further studies involving animal models or clinical trials are essential.

I suggest including information regarding the advantages and limitations of the study.

Author Response

Dear Reviewer,

We express our sincere gratitude for your constructive comments and the time you dedicated to reviewing our manuscript. We have diligently worked to enhance the quality of the manuscript in alignment with your valuable feedback. Please find our responses to your comments below:

Dear authors, the submitted manuscript is well-conceived, in-depth, informative, and scientifically structured.   However, I would like to add a few suggestions that might be implemented to modify the manuscript for better quality control.

The introduction section could be more comprehensive.

The introduction has been revised taking into account your comments.

While the study on sulforaphane and its potential interaction with NAMPT is intriguing, it's important to acknowledge certain weaknesses or limitations in the research: The conclusion that sulforaphane may activate NAMPT is based on computational predictions. Experimental evidence, such as enzymatic assays or in vitro studies, is necessary to confirm this activation and to understand the extent of its impact. The study primarily focuses on molecular interactions and may lack direct clinical evidence. To establish the practical significance of sulforaphane's interaction with NAMPT, further studies involving animal models or clinical trials are essential.

I suggest including information regarding the advantages and limitations of the study.

As reiterated in several sections of the manuscript, including the conclusion, it is important to emphasize that the data presented in this study are entirely bioinformatic and predictive. Future research should attempt to experimentally validate these results. In fact, the manuscript was submitted in a special issue focused on purely bioinformatics findings "Recent Advances in Bioinformatics: Novel Techniques, Methods, and Applications". However, in response to your suggestion, a new paragraph has been incorporated to explicitly highlight this aspect. This appears as follows in the manuscript: 

  1. Limitations of the study

The present study provides exclusively bioinformatically generated data, so it is neces-sary to emphasize the predictive nature of the results presented. As reiterated in other sections, experimental validation in future studies is imperative. Indeed, these findings could serve as a motivation for research groups around the world, inspiring them to experimentally validate our results and further investigate the mechanisms underlying the beneficial effects induced by sulforaphane.

Reviewer 3 Report

Comments and Suggestions for Authors

1. Fig. 1-3 are too vague to meet publication requirements and need to be redrawn.

2. The author should clarify the abbreviations in the abstracts.

3. Based on 3.1 and 3.2 it should be possible to identify NAMPT as the primary candidate gene, what is the role of section 3.3 in the text?

4. The author needs to analyze the gene expression levels of the 11 candidate proteins in Table 1.

5. The current manuscript needs to include some experimental verification results of NAMPT.

6. In the method, the author needs to provide a detailed description of the immortalized epithelial cells treatment.

7. The author needs to indicate the date each database was used.

8. The author needs to reorganize the text format of the supplementary file for easy reading.

Author Response

Dear Reviewer,

We express our sincere gratitude for your constructive comments and the time you dedicated to reviewing our manuscript. We have diligently worked to enhance the quality of the manuscript in alignment with your valuable feedback. Please find our responses to your comments below:

  1. Fig. 1-3 are too vague to meet publication requirements and need to be redrawn.

We believe that Figures 1 to 3 are thoroughly detailed, easily understandable, and of sufficient quality for publication, particularly in light of the feedback provided by the other two reviewers.

  1. The author should clarify the abbreviations in the abstracts.

It was corrected as you requested.

  1. Based on 3.1 and 3.2 it should be possible to identify NAMPT as the primary candidate gene, what is the role of section 3.3 in the text?

The main objective of section 3.3 is to reinforce the results of the functional enrichment analysis by providing a clear description of the members involved in each functional group and illustrating their interconnections.

  1. The author needs to analyze the gene expression levels of the 11 candidate proteins in Table 1.

This information was added and appears as follows in the manuscript:

Significantly, NAMPT, identified as a potential target of sulforaphane in this study, was shown to be overexpressed (1.03-fold increase, padj=0.042). No other proteins predicted as potential targets of sulforaphane showed significant over- or underexpression.

  1. The current manuscript needs to include some experimental verification results of NAMPT.

As reiterated in several sections of the manuscript, including the conclusion, it is important to emphasize that the data presented in this study are entirely bioinformatic and predictive. Future research should attempt to experimentally validate these results. In fact, the manuscript was submitted in a special issue focused on purely bioinformatics findings "Recent Advances in Bioinformatics: Novel Techniques, Methods, and Applications". However, in response to your suggestion, a new paragraph has been incorporated to explicitly highlight this aspect. This appears as follows in the manuscript:

  1. Limitations of the study

The present study provides exclusively bioinformatically generated data, so it is neces-sary to emphasize the predictive nature of the results presented. As reiterated in other sections, experimental validation in future studies is imperative. Indeed, these findings could serve as a motivation for research groups around the world, inspiring them to experimentally validate our results and further investigate the mechanisms underlying the beneficial effects induced by sulforaphane.

  1. In the method, the author needs to provide a detailed description of the immortalized epithelial cells treatment.

Following your suggestion, this information has been added and is now incorporated into the manuscript as follows:

The data were from WT 9-12 cells subjected to treatment with 0.1% DMSO or 10 μM sulforaphane for 24 h, followed by sequencing on the Illumina NovaSeq 6000 platform. WT 9-12 cells are epithelial cell lines that have been immortalized with SV40 large T antigen.

  1. The author needs to indicate the date each database was used.

This information was added where appropriate.

  1. The author needs to reorganize the text format of the supplementary file for easy reading

The supplementary tables were improved as you suggested

Round 2

Reviewer 3 Report

Comments and Suggestions for Authors

I have checked the updated PDF. This manuscript can be accepted in the present form.